# Influence of Sporulation Temperature on Germination and Growth of *B. weihenstephanensis* Strains in Specific Nutrients and in an Extended Shelf-Life Refrigerated Matrix Under Commercial Pasteurization and Storage Conditions

**DOI:** 10.3390/foods13213434

**Published:** 2024-10-28

**Authors:** Víctor Freire, Lina Casañas, Luis Laborda, Santiago Condón, Elisa Gayán

**Affiliations:** Department of Animal Production and Food Science, AgriFood Institute of Aragon (IA2), Faculty of Veterinary, University of Zaragoza-CITA, Miguel Servet 177, 50013 Zaragoza, Spain; vfreire@unizar.es (V.F.); lcasanas@unizar.es (L.C.); scondon@unizar.es (S.C.)

**Keywords:** *Bacillus weihenstephanensis*, bacterial spores, sporulation temperature, germination, heat resistance, extended shelf-life refrigerated foods

## Abstract

Extended shelf-life (ESL) refrigerated ready-to-eat foods are thermally pasteurized to ensure food safety and stability. However, surviving psychrotrophic *Bacillus cereus* spores can still pose a challenge. Studies predicting their behavior often overlook sporulation conditions. This study investigated the effect of sporulation temperature on germination of three *Bacillus weihenstephanensis* strains in specific nutrients (inosine and/or amino acids) with or without prior heat activation (80 °C, 10 min). Sporulation temperature variably affected germination, with stronger effects in moderately responsive strains and nutrients. Heat activation strongly stimulated germination, particularly in nutrients with poorer responses, mitigating differences induced by sporulation temperature. The influence of sporulation temperature on germination and growth in an ESL matrix at refrigeration temperatures (4 °C or 8 °C) in vacuum packaging after heat activation or commercial pasteurization (90 °C, 10 min) was also studied. The latter treatment increased germination rates of surviving spores; however, some strains suffered damage and lost viability upon germination at 4 °C but recovered and grew at 8 °C. These findings highlight the need to account for variability in spore recovery and outgrowth during quantitative risk assessments for psychrotrophic *B. cereus* in ESL foods.

## 1. Introduction

The demand for refrigerated ready-to-eat foods with extended shelf-life (hereafter referred to as ESL foods) has been increasingly growing due to modern lifestyles and consumer preferences for minimally processed products [1]. However, contaminating bacterial spores pose a challenge to ensure their safety and extended shelf-life [2,3]. Sporulation is one of the most enduring strategies used by some Firmicutes bacteria to survive adverse environmental conditions. It ultimately results in the release of a spore from the mother vegetative cell, which is metabolically inactive but capable of withstanding external stresses for prolonged periods, including food preservation methods. The higher resistance of spores compared to vegetative cells is mainly due to their low water content and the presence of specific spore molecules in their core, making them appear microscopically as refringent cells (Figure 1), and the enclosure of the core by multiple layers of unique structure and composition. When favorable conditions return, the spores can germinate, resulting in vegetative cells capable of resuming growth [4].

ESL products are typically subjected to thermal pasteurization at 90 °C for 10 min or to an equivalent treatment that achieves a 6-log reduction of non-proteolytic *Clostridium botulinum* spores due to the severity of illness caused by their toxins [5,6]. The growth of surviving spores is controlled by refrigerated storage, with a recommended temperature of 8 °C or lower, often complemented by other mild barriers such as reduced pH, water activity (*a_w_*), controlled atmosphere, and the presence of antimicrobials [2,7]. However, these barriers cannot completely prevent the germination and growth of certain psychrotrophic spores, which limits the shelf-life of ESL foods to about 10 to >40 days [2,8].

Psychrotrophic *Bacillus cereus* spores, especially from strains belonging to phylogenetic group VI as proposed by Guinebretière et al. [9], are commonly encountered in ESL products [10,11,12]. Although group VI strains produce the most heat-sensitive spores among *B. cereus sensu lato*, there is considerable intraspecific variation, with some spores exhibiting greater heat resistance than non-proteolytic *C. botulinum* spores [3,8,13,14]. Thus, the recommended pasteurization treatment can render surviving spores with sublethal injuries and even induce germination [15]. Most group VI strains can grow at temperatures as low as 4 °C [16,17] and can cause spoilage at cell concentrations between 10^5^ CFU/mL and 10^7^ CFU/mL [12,18]. In addition, most psychrotrophic strains can produce at least one enterotoxin that causes diarrheal disease (NheA and/or HblA), and some of them can also synthesize the cereulide toxin that leads to emetic disease [7,17]. *B. cereus* vegetative cells or spores at concentrations as low as 10^3^ CFU/g have been associated with both diarrheal and emetic cases [19,20]. However, the pathogenicity of psychrotrophic strains is lower than that of mesophilic strains [7,19,20]. Exposure assessments for psychrotrophic *B. cereus* in pasteurized chilled foods indicate that the risk is relatively low in properly stored products, and no cases of food poisoning have been associated with ESL products [7,19]. However, despite the likelihood of underreporting due to the self-limiting symptomatology of *B. cereus* diseases, there has been an increasing trend in the number of notifications of foodborne outbreaks and individual cases involving this bacterium over the past decade [7,17].

Several models have been proposed to predict germination, growth, and toxin production in psychrotrophic *B. cereus* spores, taking into account factors such as intensity of heat treatment, storage conditions (including temperature, food composition, and oxygen presence), and strain variability [15,21,22,23,24,25,26,27]. However, accurate prediction remains challenging due to considerable uncertainty in spore properties. Heat resistance and germination of bacterial spores are affected not only by strain diversity but also by sporulation conditions, with temperature being one of the most influential factors [28,29,30,31]. In fact, variations in sporulation conditions, such as temperature and medium composition, can lead to similar or even greater variability in spore heat resistance compared to inherent differences among strains [29,32]. This variability is critical to consider when assessing food safety and stability, given that spores form in niches with diverse and sometimes fluctuating temperatures. *B. cereus* spores that contaminate food typically originate from soil, animal and insect gastrointestinal tracts, plant surfaces, the rhizosphere, and food processing facilities, especially from biofilms that survive cleaning processes [33,34,35].

The impact of sporulation temperature on the heat resistance and germination of *B. cereus* spores has been extensively studied [31,36,37,38]. While it is widely acknowledged that heat resistance decreases with sporulation temperature, data on the magnitude and direction of its effect on germination kinetics remain conflicting [28,31,37,38]. Furthermore, most studies investigating the effect of sporulation temperature on *B. cereus* spore germination have been conducted using well-defined nutrient media (containing nucleosides and/or amino acids) at optimal growth temperatures, and generally after spores have been exposed to a thermal activation treatment to enhance and synchronize nutrient-induced germination ([39,40]). Therefore, extrapolating these findings to complex ESL foods, which generally undergo higher-intensity pasteurization treatments compared to activation [8] and are stored at refrigerated temperatures, often alongside other preservation stresses, presents significant challenges.

The primary objective of the present study was to investigate the impact of sporulation temperature on the germination of spores derived from three strains of psychrotrophic *B. weihenstephanensis*, including both pathogenic and food isolates. This was investigated over a wide range of specific nutrients and within a complex medium designed to simulate an ESL matrix. In addition, the study aimed to evaluate the effect of prior heat activation (at 80 °C for 10 min) or pasteurization treatment (at 90 °C for 10 min) on germination. Furthermore, we provide new insights into the influence of sporulation temperature on germination and subsequent growth in the ESL matrix subjected to pasteurization treatment and stored in vacuum packaging at ideal (4 °C) and maximum recommended (8 °C) temperatures.

## 2. Materials and Methods

This study was carried out using *B. weihenstephanensis* strains WSBC 10204 and WSBC 10202, provided by the *Bacillus* Genetic Stock Center (Columbus, OH, USA). The former corresponds to the type strain of the *B. weihenstephanensis* group, while the latter was selected for its high HblD and NheB activity, corresponding to hemolytic and non-hemolytic toxins, respectively [41]. Additionally, a *B. weihenstephanensis* food isolate [42], denominated as SC in this work, was included. All strains were stored at −80 °C in cryovials of Nutrient Broth No. 2 (NB; Oxoid, Basingstoke, UK) supplemented with 25% glycerol (PanReac, Darmstadt, Germany). Cells were revitalized on Nutrient Agar (Oxoid) supplemented with 0.6% Yeast Extract (Oxoid) (NAYE) at 25 °C for 24 h.

For sporulation, a preculture was prepared by inoculating a single colony into a 60 mL flask containing 10 mL of NB and incubated at 25 °C for 12 h with orbital shaking (250 rpm). Then, 20 µL of the preculture was inoculated into a 250 mL flask containing 20 mL of the sporulation medium consisting of 8 g/L NB, 4.5 mM CaCl_2_ (Sigma-Aldrich, St. Louis, MO, USA), 0.33 mM MnSO_4_ (Carlo Erba, Barcelona, Spain), and 1.05 mM MgCl_2_ (Sigma-Aldrich). Subsequently, these cultures were incubated at 10 °C, 15 °C, 20 °C, 25 °C, 30 °C, or 37 °C with orbital shaking (250 rpm). Sporulation yield was monitored over time by counting plates before and after heat treatment at 75 °C for 10 min. The optimum sporulation temperature (T_opt_) for each strain was set at the value that resulted in the fastest sporulation rate and highest sporulation yield, while the lower- and upper-temperature boundaries for sporulation (T_min_ and T_max_, respectively) were set at the value just prior to that at which sporulation yield started to decrease. To ensure that all spore samples were harvested at a similar age of maturation, sporulation for each strain and incubation temperature was concluded when two equal plate counts were obtained on two consecutive days. Table 1 compiles T_min_, T_opt_, and T_max_, along with the corresponding incubation times and spore yields achieved for each strain.

For spore purification, cultures were centrifuged (3345× *g*, 20 min, 4 °C; Megafuge 1.0, Osterode, Germany), and the pellets were washed three times with 100 mM potassium phosphate buffer (pH 7.4) supplemented with 0.1% Tween 80 (Biolife, Milan, Italy) to prevent spore aggregation [43,44]. The spores were then washed three times with the potassium phosphate buffer but supplemented with 0.01% Tween 80. Finally, spores were purified by gradient buoyant centrifugation using Nycodenz^®^ (Axis-Shield Ltd., Dundee, UK), as previously described by Ghosh and Setlow [45]. Spore purity (>98% phase-bright free spores) and disaggregation were verified by phase-contrast microscopy (Nikon Eclipse E400, Tokyo, Japan). Spore suspensions were stored at −20 °C until use. To assess variability, we obtained three spore populations per sporulation temperature and strain on independent days.

### 2.1. Germination Kinetics at Optimal Growth Conditions

Germination kinetics were assessed by monitoring the reduction in optical density at 600 nm (OD_600_), resulting from the release of dipicolinic acid (DPA) and rehydration of spores. Spore suspensions were adjusted to an OD_600_ of 1.0 (approximately 2 × 10^8^ spores/mL) in potassium phosphate buffer (100 mM, pH 7.4) containing 0.01% Tween 80. Subsequently, samples were diluted in a 1-to-2 ration in different germination media containing a final concentration of inosine (20 mM, Sigma-Aldrich), L-alanine (100 mM, Sigma-Aldrich), L-glutamine (50 mM, Sigma-Aldrich), L-glycine (50 mM, AnalaR Normapur), or combinations of the nucleoside with each amino acid (inosine (12.5 mM) and L-alanine (25 mM), inosine (0.1 mM) and L-glutamine (20 mM), or inosine (0.1 mM) and L-glycine (20 mM)). These nutrient concentrations were selected according to the existing literature, with some variations [31,46,47,48]. As an ESL-like matrix, we used NBYE supplemented with 1.5% NaCl (PanReac), adjusted to pH 6.2 with 2 M HCl (PanReac) and to *a_w_* 0.985 with glycerol (PanReac). pH measurements were conducted with a pH meter (Selecta, pH meter Basic 20+, Barcelona, Spain), and *a_w_* measurements were performed at room temperature using a Decagon CX-1 (Decagon Devices Inc., Pullman, WA, USA). Additionally, the ESL matrix was supplemented with a final concentration of 4 µg/mL chloramphenicol, to avoid the interference of growth of the first germinated spores. The presence of the antibiotic did not affect the germination kinetics. When indicated, spores were heat-activated prior to inoculation into the germination media at 80 °C for 10, as described below.

At 25 °C, OD_600_ was measured using a multiwell plate reader (CLARIOstar Plus, BMG, Ortenberg, Germany), which automatically recorded data every 3 min. Each reading was preceded by 30 s of shaking to prevent spore sedimentation. Germination curves were generated by calculating the percentage reduction in OD_600_ (OD_t_/OD_i_ × 100, where OD_i_ is the initial value and OD_t_ represents the measurement at subsequent incubation times). To estimate kinetic parameters, germination curves were fitted to the One-Phase Decay equation (Equation (1)) using GraphPad PRISM 5.0 (GraphPad Software Inc., San Diego, CA, USA). This model characterizes germination curves using two parameters: k (min^−1^), the germination rate constant, and the plateau, which represents the percentage decrease in OD_600_ at infinite time. The fit quality was assessed by calculating R^2^ and the root mean square error (RMSE). Curves where the OD_600_ reduction fell below 15% under environmental conditions were excluded from modeling.
OD_600_ fall (%) = (100 − *plateau*) *e*^(−*kt*)^ + *plateau*(1)

For a more accurate comparison of germination efficiency, the percentage of germinated spores at the end of the assays was determined by phase-contrast microscopy. A total of 100 to 150 individuals per sample were examined and categorized as either dormant (phase-bright cells) or germinated spores (phase-dark and -gray cells). The upper and lower limit of quantification for germinated spores was approximately 6.0% and 98.0%, respectively.

For each nutrient and environmental condition, we obtained three germination curves from different biological replicates.

### 2.2. Heat Treatment and Determination of Viability

Heat treatments were performed in a VeritiPro thermal cycler (Thermo Fisher Scientific, Waltham, MA, USA) using PCR tubes containing 65 µL per sample. To activate germination, spores were exposed to 80 °C for 10 min [49]. To simulate a commercial pasteurization treatment for controlling non-proteolytic *C. botulinum* spores, samples were treated at 90 °C for 10 min in the ESL matrix ([5,6]). Following treatment, samples were serially diluted in Maximum Recovery Diluent (MRD, Sigma-Aldrich) and pour-plated in NAYE. After incubation at 25 °C for 24 h, plate counts were obtained using an automatic colony counting system by image analysis. Survival was calculated as the difference between logarithm of *N*_t_ and *N*_0_ (log (*N*_t_/*N*_0_)), representing the number of CFU/mL after and before treatment, respectively. The limit of quantification was 300 CFU/mL.

### 2.3. Germination and Growth Kinetics in the ESL Matrix Under Industrial Conditions

To evaluate germination and growth kinetics under industrial processing and storage conditions, spores were suspended in the ESL matrix to a concentration that rendered approximately 4 log CFU/mL after heat treatment. Two samples per strain and sporulation temperature were prepared, one supplemented with chloramphenicol to assess germination kinetics and one without the antibiotic to determine growth kinetics. Samples were immediately treated at 90 °C for 10 min or 80 °C for 10 min and then transferred to a 96-well PCR plate, which was sealed with a gas-permeable film (Thermo Fisher Scientific) and then vacuum-packed (Qxmcov, Xixiang, China). Sample preparation was performed in a 4 °C room to prevent germination prior to the start of the assay. Samples were then incubated at 4 °C or 8 °C for up to 40 days. The percentage of germination was determined by the difference in plate counts after heat treatment at 75 °C for 10 min between time zero and various interval times. The limit of quantification was 300 CFU/mL. Growth kinetics in chloramphenicol-free samples were determined by plate counting without prior heat exposure. In this case, viability was determined by spotting 5 μL drops of different dilutions onto NAYE, as previously described [50]. After incubation (25 °C, 24 h), spots containing between 5 and 50 colonies were manually counted. The limit of detection when using this plating technique was 200 CFU/mL.

### 2.4. Statistical Analysis

One-way ANOVA with Tukey’s post hoc and unpaired parametric *t*-test were performed using GraphPad PRISM 5.0. Differences were considered statistically significant when *p* was ≤0.05. Data in the figures correspond to averages and standard deviations calculated from three biological replicates, which were obtained on different working days.

## 3. Results and Discussion

### 3.1. Influence of Sporulation Temperature on Germination Kinetics of B. weihenstephanensis Spores in Specific Nutrients

To elucidate the effect of sporulation temperature, the germination kinetics of *B. weihenstephanensis* spores from three different strains produced at the minimum (T_min_), optimum (T_opt_), and maximum (T_max_) sporulation temperatures (Table 1) were studied in a wide range of known nutrients: inosine, amino acids (L-alanine, L-glutamine, and L-glycine), and the combination of each amino acid with the nucleoside. Germination was tested both before and after thermal treatment of spores at 80 °C for 10 min (Appendix A, respectively), as heat activation can influence the effect of sporulation temperature on germination [29,31]. Table 2 shows the germination efficiency after 4 h kinetics determined by phase-contrast microscopy. The germination rate obtained from fitting the curves to the One-Phase Decay model (Equation (1)) is presented in Table 3, and the corresponding R^2^ and RMSE values are compiled in Appendix A.

In non-heat activated spores, exposure to inosine (20 mM) induced germination (>6%) in WSBC 10202 and SC spores, especially in those produced at the highest temperature, but not in any of the WSBC 10204 populations (Table 2, Appendix A). Both WSBC 10202 and SC populations prepared at T_max_ exhibited approximately 2.4-fold higher (*p* ≤ 0.05) germination efficiency than the corresponding spores prepared at T_opt_, while spores incubated at T_min_ germinated to a similar (*p* > 0.05) extent compared to the latter (Table 2). The response of all strains to inosine improved markedly after exposure to the thermal activation treatment (Appendix A). Heat-activated WSBC 10202 spores produced at T_min_ germinated up to about 94%, and those sporulated at T_opt_ and T_max_ reached ≥98% of germinated cells (Table 2). WSBC 10202 spores prepared at T_max_ remained the fastest (*p* ≤ 0.05) germinating population after heat activation compared to spores from T_opt_ and T_min_ (Table 3). On the other hand, heat activation could induce germination in about 86% of all WSBC 10204 populations with a similar (*p* > 0.05) germination rate, regardless of sporulation temperature (Table 2 and Table 3). Thermal activation increased (*p* ≤ 0.05) the germination efficiency and rate of SC spores cultured at the optimum temperature by at least 2.5- and 3.8-fold, respectively. In addition, heat treatment improved (*p* ≤ 0.05) the germination rate of spores produced at T_min_ and, to a lesser extent, that of spores formed at T_max_ (4.9-fold and 2.7-fold, respectively; Table 3), but did not affect (*p* > 0.05) their germination efficiencies (Table 2). Therefore, the germination kinetic parameters of SC spores prepared at T_opt_ and T_max_ were similar (*p* > 0.05) after heat activation (Table 2 and Table 3).

The three amino acids under investigation were unable to induce germination in any non-activated *B. weihenstephanensis* spores when tested independently (Table 2, Appendix A). Only SC spores produced at T_opt_ and WSBC 10204 spores produced at T_max_ showed significant germination efficiency (>6.0%, *p* ≤ 0.05) after 4 h exposure to L-alanine (100 mM) and L-glutamine (50 mM), respectively (Table 2). Upon activation, all spores responded to L-alanine (100 mM) and all WSBC 10204 populations also responded to L-glutamine (Table 2, Appendix A). The germination of heat-activated spores induced by L-alanine was significantly influenced by sporulation temperature in strains WSBC 10202 and SC. In both cases, spores produced at T_min_ were the least proficient in germination, especially strain SC. The germination efficiency of SC spores prepared at both T_opt_ and T_max_ was approximately ≥7.0-fold higher (*p* ≤ 0.05) than that of spores from T_min_ (Table 2). In contrast, the three heat-activated populations of WSBC 10204, the strain with the highest germination efficiency in L-alanine, showed comparable (*p* > 0.05) fractions of germinated spores and similar rates of OD_600_ decrease, irrespective of sporulation temperature (Table 2 and Table 3).

It is well known that the addition of reduced concentrations of inosine as a co-germinant with certain L-amino acids can enhance germination due to cooperativity between different types of germinant receptors (GRs), the extent of which varies among strains and with the application of an activation treatment [48,51]. This study shows that cooperativity between GRs also varies with sporulation temperature. As previously reported [31,47,48], the combination of inosine (12.5 mM) and L-alanine (25 mM) proved to be the most effective nutrient in non-activated spores, showing the highest (*p* ≤ 0.05) *k* values and percentages of germination across all strains and sporulation temperatures (Table 2 and Table 3). In strains WSBC 10202 and SC, the combination of inosine (12.5 mM) and L-alanine (25 mM) increased (*p* ≤ 0.05) the germination efficiency and rate observed with inosine (20 mM) or L-alanine (100 mM) alone in non-activated spores from T_opt_ and T_min_ to a greater extent than in spores produced at the highest temperature (Table 2 and Table 3). Consequently, unlike inosine alone, non-activated WSBC 10202 and SC spores prepared at T_opt_ germinated with the same (*p* > 0.05) or even higher (*p* ≤ 0.05) efficiency and rate than those sporulated at T_max_ in the combination of inosine and L-alanine (Table 2 and Table 3). However, spores produced at T_min_ germinated less efficiently and more slowly (*p* ≤ 0.05) than the other two populations (Table 2). More strikingly, the three WSBC 10204 populations, which were initially unresponsive to inosine and L-alanine alone, equally (*p* > 0.05), achieved an average germination efficiency of 95% and exhibited a similar (*p* > 0.05) germination rate (Table 2 and Table 3). Heat activation could only largely improve (*p* ≤ 0.05) the germination efficiency induced by inosine plus L-alanine in WSBC 10202 spores prepared at T_min_ (Table 2), but their *k* values were still 2.6- and 5.0-fold lower (*p* ≤ 0.05) than those from T_opt_ and T_max_, respectively (Table 3). Conversely, heat activation impaired the germination of SC spores produced at T_min_ and T_opt_: the germination rate decreased (*p* ≤ 0.05) 4.1- and 6.8-fold, respectively, and the percentage of germination of T_min_ spores decreased (*p* ≤ 0.05) 2.6-fold, but it did not affect (*p* > 0.05) the germination kinetic parameters of spores from T_max_ (Table 2 and Table 3).

The combination of inosine (0.1 mM) and L-glutamine (20 mM) improved the germination of non-activated spores compared to the individual germinants (20 mM inosine or 50 mM L-glutamine) only in strain WSBC 10204 (Table 2 and Table 3, Appendix A). Non-activated WSBC10204 spores produced at T_max_ exhibited a significantly higher (*p* ≤ 0.05) germination efficiency and rate than those prepared at the optimum temperature (Table 2), while spores from T_min_ showed a lower (*p* ≤ 0.05) percentage of phase-dark spores compared to those sporulated at T_opt_ (Table 3). However, heat-activated WSBC 10204 spores germinated to ≥98% with no influence (*p* > 0.05) of sporulation temperature on their kinetics (Table 2). In contrast, the combination of inosine and L-glutamine inhibited germination of all populations of non-activated WSBC 10202 spores, although they were responsive to inosine alone (Table 2). The percentage of germination in all WSBC 10202 populations increased to >90% when heat-activated (Table 2), but spores produced at T_min_ exhibited lower (*p* ≤ 0.05) germination efficiency and rate than those from T_opt_ and T_max_ (Table 2 and Table 3). On the other hand, heat-activated WSBC 10202 spores prepared at T_max_ showed higher (*p* ≤ 0.05) *k* values than those from T_opt_ (Table 3). The combination of inosine and L-glutamine also decreased (*p* ≤ 0.05) the germination efficiency of non-activated SC spores compared to inosine alone, especially those produced at T_opt_ (Table 2). Heat activation enhanced germination of the three populations in the mixture, but spores produced at the lowest temperature remained the least fit population (Table 2 and Table 3).

The combination of inosine (0.1 mM) and L-glycine (20 mM) increased (*p* ≤ 0.05) the germination efficiency and the rate of non-activated spores of strain SC prepared at the optimum temperature and, to a lesser extent, those produced at the maximum temperature compared to the single germinants (20 mM inosine or 50 mM L-glycine), so that both populations showed similar (*p* > 0.05) values (Table 2 and Table 3). The germination rate of both populations was further enhanced (*p* ≤ 0.05) after heat activation (Table 3). However, the combination of inosine and L-glycine did not improve the germination of either non-activated or heat-activated SC spores produced at the suboptimal temperature compared to inosine alone (Table 2 and Table 3). In WSBC 10202, the mixture of inosine and L-glycine enhanced only inosine-induced germination after heat exposure, with T_min_ spores showing the lowest (*p* ≤ 0.05) germination rate (Table 2 and Table 3). The same occurred in strain WSBC 10204, but both germination efficiency and rate increased to the same (*p* > 0.05) level, regardless of sporulation temperature (Table 2 and Table 3).

To sum up, variations in sporulation temperature significantly altered germination kinetics of non-activated spores, with the greatest variation observed in strain SC. Spores prepared at the highest sporulation temperature generally germinated faster and to a higher extent than those produced at the optimal temperature, while in specific cases (strain WSBC 10204 in the combination of inosine and L-glutamine and SC in the combination of inosine and L-glycine) spores produced at the lowest temperature were less responsive. In the most effective germinant (inosine plus L-alanine), variations in sporulation temperature only affected the germination efficiency and rate of WSBC 10202 and SC spores but to a lesser extent compared to other nutrients, with spores produced at T_min_ generally presenting lower germination fitness.

The changes in spore composition and structure induced by sporulation temperature, which are responsible for the different germination dynamics, are not yet well understood. In *B. subtilis* PS832, the lower germination efficiency of spores produced at 23 °C compared to those sporulated at 37 °C was positively correlated with the number of GRs [52]. On the other hand, it has been found that the expression levels of genes encoding the GRs in *B. weihenstephanensis* KBAB4 were not significantly different at various sporulation temperatures, although it was not confirmed that the same occurred at the protein level [31]. In addition, changes in DPA content and coat properties caused by variations in sporulation temperature have also been associated with different germination patterns [36,38,53].

It has been well described that the required temperature and time for heat activation varies with the type of nutrient and strain [39,54,55,56]. In agreement with this, we observed that heat activation promoted germination in certain nutrients, particularly inosine, L-alanine, and combinations of the nucleoside with any of the amino acids, especially in strains and sporulation temperatures with poorer responses. Differences in germination of heat-activated WSBC 10202 spores induced by changes in sporulation temperature were attenuated compared to non-activated populations, but spores produced at the suboptimal temperature often still had lower germination efficiency and/or germination rate than those from higher temperatures. Furthermore, germination of heat-activated WSBC 10204 spores was unaffected by changes in sporulation temperature across all nutrients. An exception was SC spores, where heat activation did not affect, or even impaired (in inosine combined with L-alanine) germination of spores produced at the suboptimal temperature, thus broadening variability in germination kinetics caused by changes in sporulation temperature. Consistent with this, other authors have observed that a heat activation treatment that improved germination of KBAB4 spores produced at 30 °C and 20 °C in inosine and its combination with L-alanine had no effect on spores cultured at 12 °C [31]. The reason why heat activation did not work in SC spores produced at T_min_ is unknown. Although we verified that the heat activation treatment did not cause a significant decrease in viability, it may cause sublethal damage to some essential components of the germination pathway, because there is a fine line between the intensity of heat to activate or damage the germination and outgrowth machinery [39].

Heat activation is believed to enhance and synchronize nutrient-induced germination, either directly by altering the structure and affinity of GRs or indirectly by affecting their surrounding environment [56,57]. On the other hand, other changes have been observed in heat-activated spores that may contribute to their different response to nutrients compared to non-activated spores, such as the release of some DPA and coat proteins and the denaturalization of proteins such as alanine racemases that isomerize L-alanine to D-alanine, which compete with L-alanine for binding to GRs and thus inhibit germination [58,59,60].

Our results suggest that the enhancement of nutrient responsiveness induced by heat activation, by whatever mechanism, is influenced by as-yet unknown structural modifications, depending on sporulation temperature. Further investigation is required to elucidate the mechanism governing the interaction between heat activation and sporulation temperature.

### 3.2. Germination of B. weihenstephanensis Spores in an ESL Matrix

Subsequently, we evaluated the germination kinetics of WSBC 10202, WSBC 10204, and SC spores prepared at different sporulation temperatures in an ESL model at the optimal growth temperature (25 °C; Appendix A). This matrix consisted of a protein-rich growth medium (nutrient broth and yeast extract) with NaCl concentration (1.5%), pH (6.2), and *a_w_* (0.985) similar to that of a meat-based ESL product. The germination rate, along with the percentage of germination after 4 h of incubation, is presented in Table 4. Additionally, we evaluated the extent of germination at 24 h by plate count of heat-treated samples for higher resolution (Figure 2).

Although non-activated WSBC 10202 spores were able to germinate in inosine alone and in combination with certain amino acids (Appendix A), the percentage of germination of spores prepared at T_min_, T_opt_, and T_max_ in the ESL matrix remained ≤6% after 4 h of incubation (Table 4). Heat activation at 80 °C for 10 min did not promote germination in any of the three populations (Table 4), even when the incubation period was extended to 24 h (Figure 2). Other studies have also shown that certain spores of the *B. cereus* group fail to germinate in nutrient-rich media such as foods, food models, and rich growth media, even after heat activation, despite responding well to inosine combined with amino acids [31,48,49]. This has been attributed, on the one hand, to the lower concentrations of inducing nutrients encountered in these matrices compared to those used in laboratory experiments [49,61]. It has been described that the extent of germination correlates with the concentration of some nutrients within a certain range, the threshold of which varies with the strain [37,46]. On the other hand, it is possible that the presence of several types of nutrients simultaneously triggers multiple germination pathways that may negatively interfere with each other [62].

As in the case of WSBC 10202, non-activated spores of WSBC 10204 germinated to a lesser extent in the ESL matrix than in the specific germination-inducing nutrients (i.e., the combination of inosine and L-alanine or L-glutamine; Table 2 and Table 4). The germination efficiency of non-activated WSBC 10204 spores prepared at T_min_ and T_max_ reached approximately 25% after 4 h with similar (*p* > 0.05) *k* values, while the percentage of germinated spores in those produced at T_opt_ did not exceed 6% (Table 4). After 24 h, the germination of non-activated WSBC 10204 populations sporulated at T_min_ and T_opt_ increased to 80.0% and 36.9%, respectively (corresponding to 0.7 and 0.2 log cycles of germinated spores, respectively; Figure 2B). When activated (80 °C, 10 min), the germination rate of the three populations T_min_, T_opt_, and T_max_ increased (*p* ≤ 0.05), and the germination efficiency at 4 h of all reached on average 85.9% (*p* > 0.05). Further incubation of heat-activated spores increased the germination efficiency of the three populations, revealing noticeable differences induced by sporulation temperature (Figure 2B). Spores produced at both T_min_ and T_opt_ germinated to about 2.7 log units (*p* > 0.05), whereas those cultured at T_max_ showed a lower (*p* ≤ 0.05) degree of germination (1.3 log units).

In contrast to laboratory strains, SC spores produced at the optimum temperature reached approximately 92.5% of germinated spores after 4 h of incubation without prior heat exposure (Table 4). SC spores produced at T_min_ and T_max_ showed lower (*p* ≤ 0.05) germination efficiency than spores prepared at T_opt_, and the former also exhibited decreased (*p* ≤ 0.05) *k* values compared to spores from T_opt_. Heat activation impaired (*p* ≤ 0.05) the germination of spores produced at T_min_ but did not affect (*p* > 0.05) the germination rate and efficiency of those sporulated at T_opt_ and T_max_ (Table 4). Thus, as observed for specific nutrients (Table 2 and Table 3), non-activated and heat-activated SC spores produced at T_min_ followed the trend of being the most disadvantaged population in germination during the first 4 h. Both non-activated and heat-activated SC spores produced at T_min_ continued to germinate to a lesser extent (*p* ≤ 0.05) than those prepared at T_opt_ in the ESL matrix after 24 h (Figure 2C). Nevertheless, spores from T_max_ germinated less efficiently (*p* ≤ 0.05) than spores from T_opt_ and T_min_ (Figure 2C), as occurred in strain WSBC 10204 (Figure 2B). The advantage of heat activation on germination of SC spores observed during the first 4 h dissipated after 24 h.

Our results indicate that the effect of sporulation temperature on germination differs between specific nutrients and the ESL matrix, depending on the strain and prior heat treatment, and therefore the influence of this factor should be evaluated under conditions close to food environments. In addition, incubation time should be considered when evaluating differences in germination induced by sporulation temperature or other influential factors. While assessing germination for a short period of time, typically by spectrophotometry or microscopy, allows detection of germination of the fraction of the population with the highest germination fitness, albeit a very small one, extending the incubation time and assessing germination by plate count for higher resolution also considers changes in germination rate from slow-germinating subpopulations.

### 3.3. Impact of Heat Pasteurization on the Viability and Germination Kinetics of B. weihenstephanensis Spores in an ESL Matrix

We first evaluated the lethality of the recommended pasteurization treatment of 90 °C for 10 min on *B. weihenstephanensis* spores sporulated at different temperatures in the ESL matrix. As shown in Figure 3, in strains WSBC 10202 and SC, the survival of spores produced at T_min_ fell beyond the limit of quantification (≥6.0 log reductions), while there were no statistically significant differences (*p* > 0.05) between the inactivation of spores prepared at the optimal and supra-optimal sporulation temperature. Both WSBC 10202 populations cultured at T_opt_ and T_max_ were 2.1 and 1.4 log cycles more resistant (*p* ≤ 0.05) than those of strain SC.

As widely reported, there is a large variability in the inherent heat resistance of psychrotrophic *B. cereus* spores [7,14]. In addition, sporulation temperature modulates spore resistance properties [28,31,63,64]. Generally, decreasing the sporulation temperature from the optimum decreases the heat resistance, while the effect of the supra-optimal sporulation temperature depends on the strain and the temperature range [28,31,38,64,65]. The higher sensitivity of spores produced at the lowest temperature could be observed in strains WSBC 10202 and SC, and it is likely to exist also in strain WSBC 10204, but the low lethality of the treatment on these spores hindered the evaluation of the effect of sporulation temperature. Our results reinforce the idea that certain psychotropic *B. cereus* strains can readily survive conventional thermal pasteurization applied to ESL products and should therefore be considered as potential target pathogens for establishing thermal processing conditions [7].

We then evaluated the germination of spores treated at 90 °C for 10 min in the ESL matrix at the optimal growth temperature (25 °C; Appendix A). As it has been reported that spores with higher heat resistance require a higher intensity of thermal treatment to activate germination [55,66], we focused on spore populations whose inactivation did not exceed 1.5 log reductions (WSBC 10202 spores produced at T_opt_ and WSBC 10204 spores produced at T_min_, T_opt_, and T_max_). The germination rate and efficiency for the first 4 h are shown in Table 4, while the extent of germination after 24 h (as determined by plate counts) is included in Figure 2. In contrast to the 80 °C treatment, a more intense heat treatment (90 °C, 10 min) induced germination of WSBC 10202 spores, reaching about 21.8% germination after 4 h (Table 4) and further increasing to approximately 90.0% after 24 h of incubation (Figure 2A). In strain WSBC 10204, commercial pasteurization (90 °C, 10 min) increased the germination efficiency of spores produced at T_min_, T_opt_, and T_max_ to the same level (*p* > 0.05) as the 80 °C treatment after 4 h of incubation, with similar (*p* > 0.05) or even lower (*p* ≤ 0.05) germination rate (Table 4). Similarly, the extent of germination of the three populations treated at 90 °C after 24 h increased to the same level (*p* > 0.05) as when activated at 80 °C, resulting in spores produced at T_max_ germinating approximately 0.6 and 1.3 log units lower (*p* ≤ 0.05) than those from T_opt_ and T_min_, respectively (Figure 2B).

Previous studies have shown that thermal activation requirements differ between laboratory media and foods or food models [15,67]. Similarly, for WSBC 10202 spores, germination in most specific nutrients (except L-glutamine and L-glycine) was enhanced by heat activation at 80 °C, but required 90 °C in the ESL matrix (Figure 2A). Our findings also evidence that the pasteurization treatment commonly applied to ESL foods can enhance germination of WSBC 10202 spores despite inactivating 90% of them. Other authors have observed that commercial treatments that cause lethal effects promote germination of a subfraction of surviving spores that can initiate outgrowth earlier than untreated spores [15,68]. However, the ability of spores to repair damage and undergo outgrowth is dependent on storage temperature [15,68]. Therefore, we investigated whether heat-treated *B. weihenstephanensis* spores would compromise the safety and stability of ESL products under commercial processing and storage conditions.

### 3.4. Germination and Growth Kinetics of B. weihenstephanensis Spores in an ESL Matrix Under Commercial Storage Conditions

The germination of heat-resistant spores (WSBC 10202 spores produced at T_opt_ and WSBC 10204 spores produced at T_min_, T_opt_, and T_max_) was evaluated after applying a commercial pasteurization treatment (90 °C, 10 min) during storage in vacuum packaging and refrigeration at an ideal temperature of 4 °C. To evaluate the damage caused by the thermal treatment at 90 °C, a set of samples was subjected to a milder treatment at 80 °C for 10 min for spore activation. Figure 4 shows the evolution of the plate count of germinated spores of WSBC 10202 produced at T_opt_ and WSBC 10204 sporulated at T_min_, T_opt_, and T_max_ during 10 days of storage.

WSBC 10202 spores treated at 80 °C germinated progressively, reaching approximately 1.3 log units by day 8 (Figure 4A). Longer incubation times did not significantly (*p* > 0.05) increase germination efficiency. When treated at 90 °C, surviving spores exhibited a germination of 0.7 log units within the first day (Figure 3B), similar (*p* > 0.05) to that observed at 25 °C (Figure 2A). The number of germinated spores increased to 1.5 log units on day 2 (Figure 4B), reaching a maximum extent of germination similar (*p* > 0.05) to that of spores treated at 80 °C by day 8 (Figure 4A). WSBC 10204 spores produced at T_min_, T_opt_, and T_max_ and exposed to 80 °C showed the highest germination efficiency on day 3, while the three populations exposed to the severe treatment reached the maximum value on day 2 (Figure 4). Thus, our results indicate that a pasteurization treatment at 90 °C increased the germination rate of WSBC 10204 spores, and especially that of WSBC 10202 spores, despite killing about 90% of the population (Figure 3), compared to a sublethal treatment at 80 °C, even when stored in a vacuum atmosphere at 4 °C (Figure 4).

Importantly, the number of germinated WSBC 10204 spores produced at T_min_ and T_opt_ after 24 h, regardless of the intensity of heat treatment applied, decreased by approximately 1.8 log units (*p* ≤ 0.05) when stored at 4 °C compared to 25 °C, whereas the germination of spores produced at T_max_ decreased by only 0.7 log units (*p* ≤ 0.05) (Figure 2B and Figure 4). Consequently, although both 80 °C- and 90 °C-treated WSBC 10204 spores produced at T_max_ germinated to a lesser extent than those from T_min_ and T_opt_ in the ESL matrix at 25 °C after 24 h, their germination efficiency did not differ (*p* > 0.05) when incubated at 4 °C. Thus, differences in germination caused by changes in sporulation temperature in the ESL matrix at 25 °C were attenuated when incubated at cold temperatures, as observed in *B. weihenstephanensis* KBAB4 spores [31]. However, the relative influence of sporulation temperature on germination kinetics at different temperatures may vary with the strain and nutrient used. In this regard, we previously observed that *B. subtilis* 168 spores produced at a supra-optimal temperature maintained their better germination fitness compared to spores produced at the optimal temperature in a complex growth medium at 37 °C and at 20 °C [30].

The growth kinetics of germinated spores of WSBC 10202 and WSBC 10204 treated at 80 °C or 90 °C during storage at 4 °C in vacuum packaging are shown in Figure 5A,B, respectively. The plate counts of all populations treated at 80 °C remained constant (*p* > 0.05) up to 20 days, except for a small decrease (*p* ≤ 0.05) of approximately 0.4 log units from day 2 to day 4 (Figure 5A). Subsequently, all WSBC 10202 and WSBC 10204 populations began to grow, reaching an average of 5.3 log CFU/mL on day 23 and 4.5 log CFU/mL on day 25, respectively. The biomass reached in the ESL matrix under refrigeration and vacuum packaging was much lower than that observed under optimal growth temperature and aerobic conditions, likely due to the stress of the combined hurdles [69,70,71]. The maximum plate counts observed were similar to the levels of psychrotrophic *B. cereus* previously reported in pasteurized foods such as milk and purees under refrigeration conditions [11,12]. Conversely, the counts of all populations treated at 90 °C decreased (*p* ≤ 0.05) by approximately 1.0 log cycle within the first 9 days (Figure 5B), much more than that observed for spores treated at 80 °C during the first 6 days (Figure 5A), and did not increase with further incubation time. Thus, although the intensive treatment induced germination at a higher rate than the 80 °C treatment, most of the germinated spores were sufficiently damaged to prevent repair and subsequent growth at 4 °C in vacuum packaging. This is consistent with previous observations of the existence of a subpopulation of severely heat-treated spores that can germinate but cannot commit to outgrowth [15,68,72,73].

Due to the significant influence of recovery conditions on heat-induced damage repair [15,74], we tested whether storage at 8 °C could affect the growth ability of spores treated at 90 °C. This temperature is commonly found in a quarter of household refrigerators in European countries and is therefore still used for risk assessment [27,75,76]. Increasing the storage temperature to 8 °C did not affect the germination pattern of any strain or sporulation temperature, and it did not prevent the decrease in survival observed in the first days of storage at 4 °C (Figure 5C). However, the plate counts of all WBSC 10202 and WBSC 10204 populations were restored to the same level as observed immediately after treatment around day 8 and day 17, respectively (Figure 5C). These findings suggest that a fraction of the damaged spores could germinate, repair the damage, and regain cultivability at 8 °C but not at 4 °C. WSBC 10202 counts increased from day 2 at 8 °C, reaching up to 5.4 log CFU/mL by day 40 (Figure 5C). This bacterial concentration is high enough to cause food spoilage and both emetic and diarrheal diseases [12,19,20,77]. Although the pathogenicity of psychrotrophic *B. cereus* is considered relatively low in properly stored products [7,19,20], it should be taken into account that some strains may sustain toxin production at 8 °C [12,78]. This reinforces the need to evaluate the influence of food components and storage temperature on the risk of toxin production [79].

In contrast, WSBC 10204 germinated cells treated at 90 °C could not resume growth at 8 °C (Figure 5C). Although WSBC 10204 spores showed higher survival after the 90 °C treatment than WSBC 10202 spores when recovered in the optimal growth medium and temperature (Figure 3), the former may suffer a higher degree of damage or have a lower repair capacity when maintained in the ESL matrix at 8 °C and in vacuum packaging. Similar results were observed in pasteurized milk, by Hanson et al. [68]. While *Bacillus* spp. spores exposed to a lethal treatment of 82 °C for 30 min remained undetectable immediately after treatment and during a 14-day period at 6 °C, damaged spores could recover and grow when incubated at 10 °C. In practice, the intensity of the heat treatment, together with the maintenance of the distribution and storage temperature, is of great importance to reduce the probability of recovery and growth of damaged *B. weihenstephanensis* spores.

## 4. Conclusions

Results obtained in this work indicate that the impact of sporulation temperature on nutrient-induced germination of *B. weihenstephanensis* spores exhibits wide variation depending on the strain, nutrient, and application of prior thermal treatment. In addition, the effect of sporulation temperature on germination observed in specific nutrients cannot be extrapolated to spore behavior in food under industrial conditions. In most strains, non-activated spores produced at the highest sporulation temperature, together with those produced at the optimal temperature when heat-activated (80 °C, 10 min), showed advantageous germination in certain amino acids, inosine and/or their combinations compared to those prepared at the suboptimal temperature during the first 4 h at 25 °C. In contrast, the former spores, especially after heat treatment, were the least responsive populations in an ESL model after 24 h at the same incubation temperature. However, the extent of germination of heat-treated spores produced at the supra-optimal temperature was less impaired when the storage temperature was reduced to 4 °C than in those produced at the optimal and suboptimal temperatures.

Most importantly for food safety, the heat treatment applied commercially to ESL products to sufficiently inactivate psychrotrophic *C. botulinum* spores (90 °C, 10 min) may stimulate germination of surviving *B. weihenstephanensis* spores even under refrigeration and anoxic conditions. Germinated cells may accumulate damage and remain undetectable by plating when stored at 4 °C but recover at 8 °C, a temperature commonly found in household refrigerators. This may contribute to the difficulty in associating food poisoning caused by psychrotrophic *B. cereus* with contaminated food, along with the self-limiting symptomatology of their diseases.

Among the three strains studied, the one with moderate heat resistance and the poorest germination response could resume growth in the ESL matrix after application of the recommended pasteurization treatment during storage at 8 °C, reaching an unsatisfactory level for food safety and stability. On the other hand, the strain producing spores with the highest heat resistance and germination fitness was able to restore viability but not proliferate, independently of the sporulation temperature. Thus, this research highlights the importance of considering not only the heat resistance and germination kinetics of heat-treated spores from different strains, but also the ability of recovery to restore viability and outgrowth under industrial conditions to improve quantitative risk assessment. Furthermore, we advise revision of the reference heat-treatment conditions for ESL to prevent germination and outgrowth of psychrotrophic *B. cereus* spores, while recommending a storage temperature below 8 °C.

## Figures and Tables

**Figure 1 foods-13-03434-f001:**
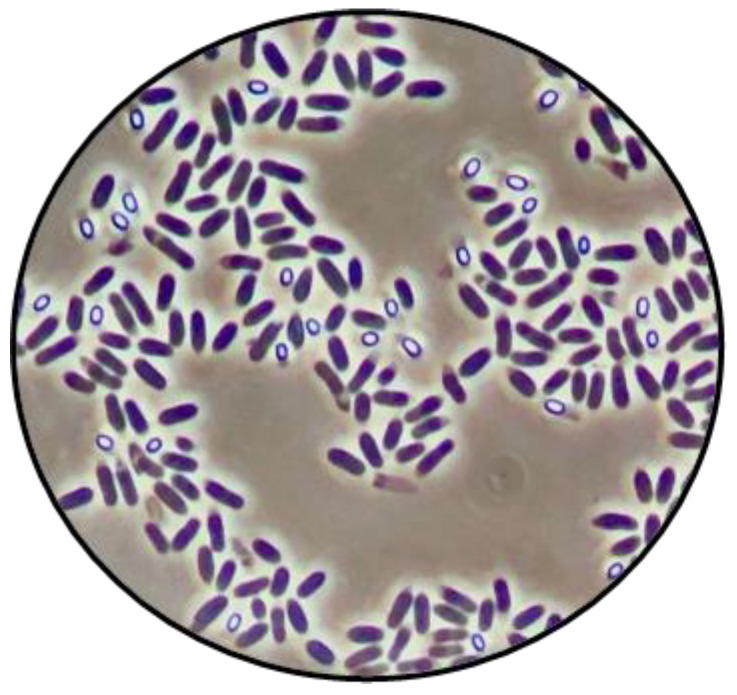
Vegetative cells (dark cells) and spores (bright cells) of *B. weihenstephanensis* WSBC 10204 observed under phase-contrast microscopy during the sporulation process.

**Figure 2 foods-13-03434-f002:**
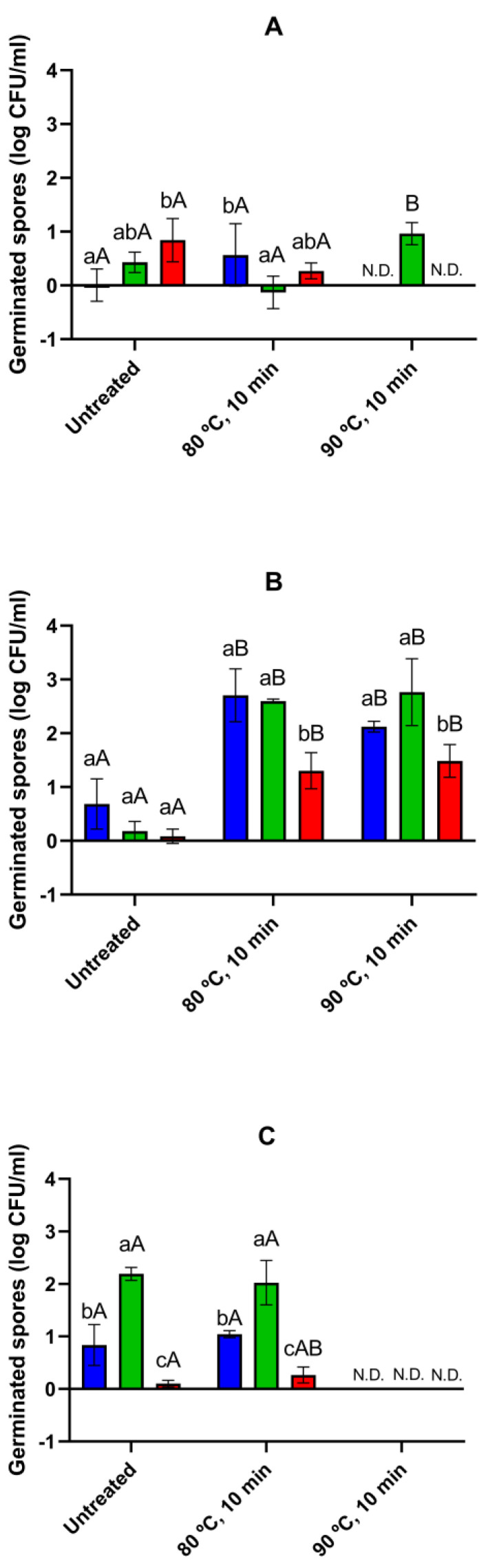
Counts (log CFU/mL) of germinated spores of strains (**A**) WSBC 10202, (**B**) WSBC 10204, and (**C**) SC produced at T_min_ (blue), T_opt_ (green), or T_max_ (red) without and with applying a prior heat treatment (80 °C, 10 min or 90 °C, 10 min) in an ESL matrix after 24 h of incubation at 25 °C. Values in the figures correspond to averages and standard deviations calculated from three biological replicates. Different lowercase letters indicate statistically significant differences (*p* ≤ 0.05) among spores produced at different sporulation temperatures within each treatment and strain. Different capital letters indicate statistically significant differences (*p* ≤ 0.05) among treatments within each strain and sporulation temperature. N.D.: not determined; germination was not tested because heat treatment caused >1.5 log reductions.

**Figure 3 foods-13-03434-f003:**
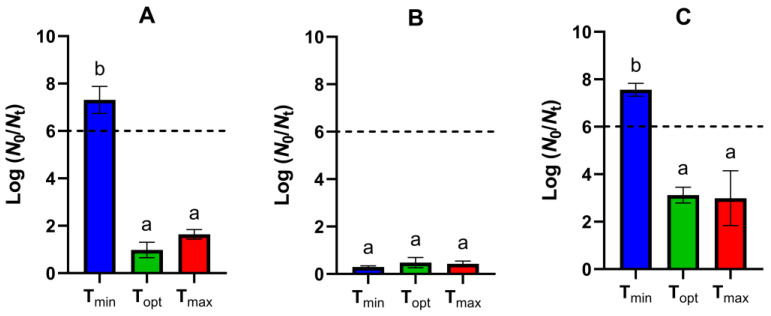
Inactivation (expressed as log (*N*_0_/*N*_t_)) of (**A**) WSBC 10202, (**B**) WSBC 10204, and (**C**) SC spores produced at T_min_ (blue), T_opt_ (green), or T_max_ (red) by heat treatment at 90 °C for 10 min in the ESL matrix. The dotted line indicates the limit of quantification (300 CFU/mL). Values in the figures correspond to averages and standard deviations calculated from three biological replicates. Different lowercase letters indicate statistically significant differences (*p* ≤ 0.05) among spores produced at different sporulation temperatures within each strain.

**Figure 4 foods-13-03434-f004:**
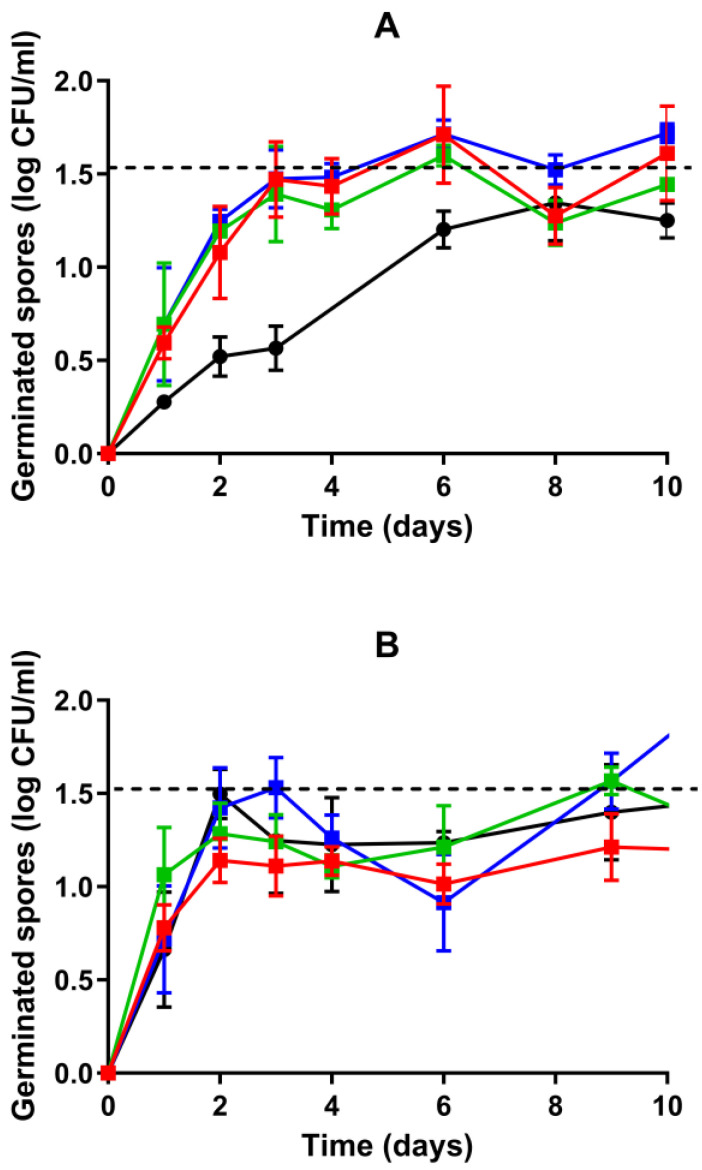
Plate counts (log CFU/mL) of germinated spores of WSBC 10202 produced at T_opt_ (●) and WSBC 10204 sporulated at T_min_ (■), T_opt_ (■), or T_max._ (■) in the ESL matrix after heat treatment at (**A**) 80 °C for 10 min or (**B**) 90 °C for 10 min during storage at 4 °C in vacuum packaging. Values in the figures correspond to averages and standard deviations calculated from three biological replicates. The dotted line indicates the limit of quantification (300 CFU/mL).

**Figure 5 foods-13-03434-f005:**
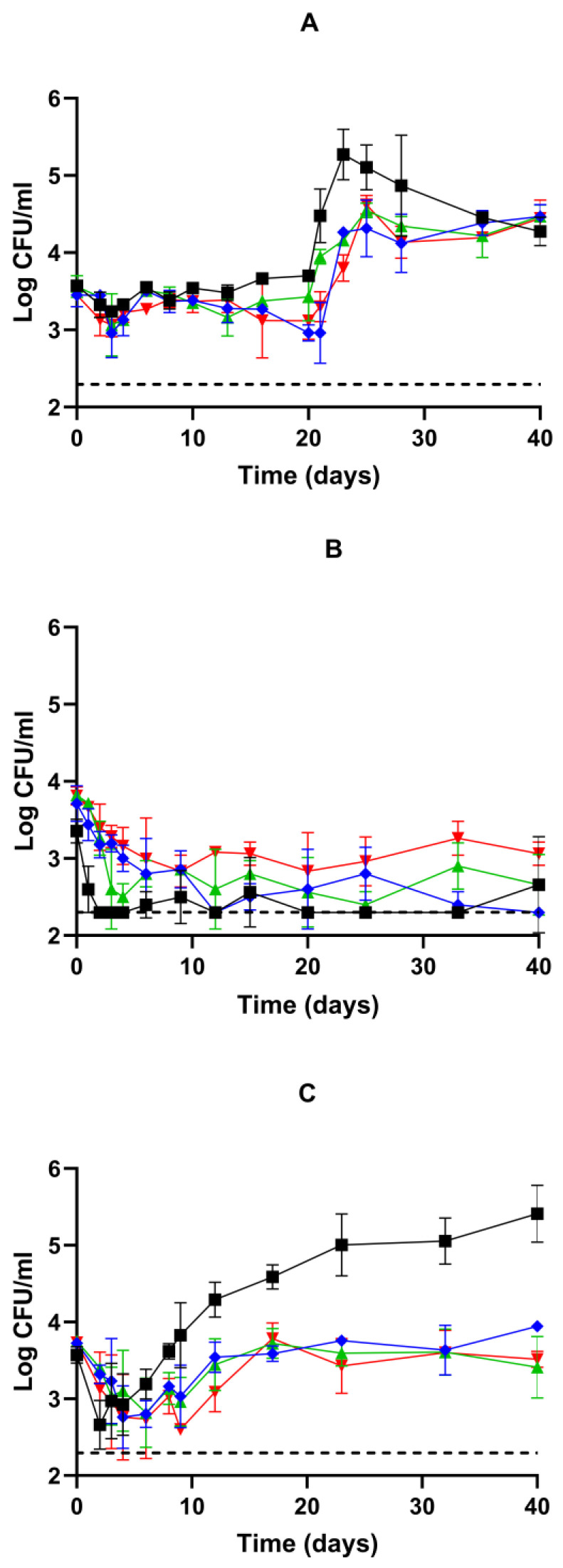
Growth kinetics of WSBC 10202 spores produced at T_opt_ (■) and WSBC 10204 sporulated at T_min_ (■), T_opt_ (■), or T_max._ (■) in an ESL matrix after heat treatment at (**A**) 80 °C for 10 min or (**B**,**C**) 90 °C for 10 min during storage at (**A**,**B**) 4 °C or (**C**) 8 °C in vacuum packaging. The dotted line indicates the limit of detection (200 CFU/mL).

**Table 1 foods-13-03434-t001:** T_min_, T_opt_, and T_max_ of sporulation, together with the corresponding incubation time and the spore yield achieved for each strain. Different letters indicate statistically significant differences (*p* ≤ 0.05) among different sporulation temperatures within each strain.

Strain	Sample	Sporulation Temperature (°C)	Incubation Time (Days)	Spore Yield (CFU/mL)
WSBC 10204	T_min_	20	6	3.48 × 10^8 a^ (1.04 × 10^8^)
	T_opt_	25	4	3.21 × 10^8 a^ (0.58 × 10^8^)
	T_max_	30	4	2.05 × 10^8 a^ (0.67 × 10^8^)
WSBC 10202	T_min_	10	15	2.97 × 10^8 a^ (0.42 × 10^8^)
	T_opt_	25	4	3.54 × 10^8 a^ (0.73 × 10^8^)
	T_max_	30	4	2.40 × 10^8 a^ (0.55 × 10^8^)
WSBC SC	T_min_	10	15	2.52 × 10^8 a^ (0.42 × 10^8^)
	T_opt_	25	4	3.26 × 10^8 a^ (0.73 × 10^8^)
	T_max_	30	4	2.21 × 10^8 a^ (0.57 × 10^8^)

**Table 2 foods-13-03434-t002:** Germination efficiency of non-activated and heat-activated (80 °C, 10 min) spores produced at different temperatures after 4 h exposure to the indicated nutrients at 25 °C. Values in brackets correspond to standard deviations of the means calculated from three biological replicates. Different lowercase letters indicate statistically significant differences (*p* ≤ 0.05) among spore populations produced at different sporulation temperatures within each strain, germinant, and activation treatment. Different capital letters indicate statistically significant differences (*p* ≤ 0.05) among different nutrients within each strain, sporulation temperature, and activation treatment. Asterisks indicate statistically significant differences (*p* ≤ 0.05) between non-activated and heat-activated samples within each strain and germinant.

		WSBC 10202		WSBC 10204		SC	
Germinant	Sporulation Temperature	Non-Activated	Activated	Non-Activated	Activated	Non-Activated	Activated
Inosine (20 mM)	T_min_	16.8 ^bA^ (1.3)	93.9 ^bA^* (0.9)	≤6.0 ^A^	86.7 ^aA^* (3.2)	29.5 ^bA^ (5.4)	24.6 ^bA^ (8.3)
	T_opt_	26.9 ^bA^ (9.0)	≥98 ^aA^*	≤6.0 ^A^	87.2 ^aA^* (5.8)	39.1 ^bA^ (8.8)	≥98 ^aA^*
	T_max_	66.8 ^aA^ (3.5)	≥98 ^aA^*	≤6.0 ^A^	85.1 ^aA^* (7.2)	86.3 ^aA^ (3.1)	88.6 ^aAD^ (10.5)
L-alanine (100 mM)	T_min_	≤6.0 ^B^	18.5 ^bB^* (5.5)	9.5 ^aA^ (6.5)	≥98.0 ^B^*	≤6.0 ^bB^	≤6.0 ^B^
	T_opt_	≤6.0 ^B^	61.0 ^aB^* (7.5)	12.7 ^aA^ (6.7)	≥98.0 ^B^*	13.6 ^aB^ (1.6)	45.6 ^aB^* (8.3)
	T_max_	≤6.0 ^B^	75.0 ^cB^* (1.9)	≤6.0 ^aA^	≥98.0 ^B^	≤6.0 ^bB^	39.2 ^aB^* (19.2)
L-glutamine (50 mM)	T_min_	≤6.0 ^B^	≤6.0 ^C^	≤6.0 ^bA^	83.8 ^aAC^* (14.7)	≤6.0 ^B^	≤6.0 ^B^
	T_opt_	≤6.0 ^B^	≤6.0 ^C^	≤6.0 ^bA^	70.5 ^aC^* (4.6)	≤6.0 ^C^	≤6.0 ^C^
	T_max_	≤6.0 ^B^	≤6.0 ^C^	23.3 ^aB^ (7.7)	85.7 ^aAB^* (11.5)	≤6.0 ^B^	≤6.0 ^C^
L-glycine (50 mM)	T_min_	≤6.0 ^B^	≤6.0 ^C^	≤6.0 ^A^	13.0 ^aD^* (9.4)	≤6.0 ^B^	≤6.0 ^B^
	T_opt_	≤6.0 ^B^	≤6.0 ^C^	≤6.0 ^A^	≤6.0 ^bD^	≤6.0 ^C^	10.9 ^C^ (5.8)
	T_max_	≤6.0 ^B^	≤6.0 ^C^	≤6.0 ^A^	≤6.0 ^bC^	≤6.0 ^B^	≤6.0 ^C^
Inosine (12.5 mM) + L-alanine (25 mM)	T_min_	63.1 ^bC^ (3.4)	96.3 ^bD^* (0.5)	91.0 ^aB^ (5.1)	≥98.0 ^B^*	88.5 ^bC^ (5.2)	33.5 ^bA^* (13.1)
	T_opt_	≥98 ^aC^	≥98 ^aA^	97.3 ^aC^ (1.0)	≥98.0 ^B^	≥98.0 ^aD^	92.7 ^aD^* (3.5)
	T_max_	≥98 ^aC^	≥98 ^aA^	96.8 ^aC^ (4.5)	≥98.0 ^B^	97.6 ^aC^ (2.2)	96.5 ^aA^ (1.0)
Inosine (0.1 mM) + L-glutamine (20 mM)	T_min_	≤6.0 ^B^	91.3 ^aAD^* (4.5)	24.7 ^cC^ (2.7)	≥98.0 ^B^*	10.0 ^aD^ (1.8)	24.1 ^bA^* (6.7)
	T_opt_	≤6.0 ^B^	≥98 ^bA^*	59.0 ^bD^ (5.2)	≥98.0 ^B^*	7.5 ^aBC^ (2.2)	≥98 ^aA^*
	T_max_	≤6.0 ^B^	≥98 ^bA^*	83.6 ^aC^ (8.4)	≥98.0 ^B^*	75.5 ^bD^ (4.5)	96.6 ^aA^* (2.0)
Inosine (0.1 mM) + L-glycine (20 mM)	T_min_	21.8 ^bA^ (9.1)	≥98 ^E^*	≤6.0 ^A^	≥98.0 ^B^	24.1 ^bA^ (1.9)	22.7 ^bA^ (0.8)
	T_opt_	23.5 ^bA^ (5.8)	≥98 ^A^*	≤6.0 ^A^	≥98.0 ^B^	91.8 ^aD^ (11.7)	97.1 ^aA^ (5.0)
	T_max_	69.6 ^aA^ (6.6)	≥98 ^A^*	≤6.0 ^A^	≥98.0 ^B^	92.4 ^aAC^ (10.7)	90.5 ^aD^ (2.8)

**Table 3 foods-13-03434-t003:** Germination rate of non-activated and heat-activated (80 °C, 10 min) spores produced at different temperatures in the indicated nutrients at 25 °C (Appendix A). Values in brackets correspond to standard deviations of the means calculated from three biological replicates. Different lowercase letters indicate statistically significant differences (*p* ≤ 0.05) among spore populations produced at different sporulation temperatures within each strain, germinant, and activation treatment. Different capital letters indicate statistically significant differences (*p* ≤ 0.05) among different nutrients within each strain, sporulation temperature, and activation treatment. Asterisks indicate statistically significant differences (*p* ≤ 0.05) between non-activated and heat-activated samples within each strain and germinant.

		WSBC 10202		WSBC 10204		SC	
Germinant	Sporulation Temperature	Non-Activated	Activated	Non-Activated	Activated	Non-Activated	Activated
Inosine (20 mM)	T_min_	N.M.	0.0371 ^aA^ (0.0062)	N.M.	0.0221 ^aA^ (0.0045)	0.0031 ^cA^ (0.0010)	0.0153 ^bA^* (0.0082)
	T_opt_	N.M.	0.0584 ^bA^ (0.0074)	N.M.	0.0173 ^aA^ (0.0039)	0.0103 ^bA^ (0.0013)	0.0394 ^aAC^* (0.0083)
	T_max_	0.0115 ^A^ (0.0021)	0.0783 ^cA^* (0.0042)	N.M.	0.0174 ^aA^ (0.0053)	0.0174 ^aA^ (0.0031)	0.0475 ^aA^* (0.0014)
L-alanine (100 mM)	T_min_	N.M.	0.0661 ^abB^ (0.0141)	N.M.	0.0615 ^aB^ (0.0052)	N.M.	N.M.
	T_opt_	N.M.	0.0351 ^bB^ (0.0008)	N.M.	0.0717 ^aB^ (0.0111)	N.M.	0.0263 ^aA^ (0.0063)
	T_max_	N.M.	0.0501 ^aB^ (0.0055)	N.M.	0.0602 ^aBE^ (0.0045)	N.M.	0.0152 ^bB^ (0.0044)
L-glutamine (50 mM)	T_min_	N.M.	N.M.	N.M.	0.0096 ^aC^ (0.0022)	N.M.	N.M.
	T_opt_	N.M.	N.M.	N.M.	0.0095 ^aC^ (0.0043)	N.M.	N.M.
	T_max_	N.M.	N.M.	N.M.	0.0097 ^aC^ (0.0024)	N.M.	N.M.
L-glycine (50 mM)	T_min_	N.M.	N.M.	N.M.	N.M.	N.M.	N.M.
	T_opt_	N.M.	N.M.	N.M.	N.M.	N.M.	N.M.
	T_max_	N.M.	N.M.	N.M.	N.M.	N.M.	N.M.
Inosine (12.5 mM) + L-alanine (25 mM)	T_min_	0.1235 ^b^ (0.0144)	0.0451 ^cAB^* (0.0091)	0.0953 ^aA^ (0.0062)	0.1001 ^aD^ (0.0097)	0.0412 ^cB^ (0.0081)	0.0101 ^bA^* (0.0073)
	T_opt_	0.2362 ^a^ (0.0214)	0.1174 ^bC^* (0.0052)	0.1183 ^aA^ (0.0281)	0.1245 ^aD^ (0.0142)	0.5623 ^aB^ (0.0513)	0.0831 ^aB^* (0.0054)
	T_max_	0.2881 ^aB^ (0.0460)	0.2252 ^aC^ (0.0314)	0.1250 ^aA^ (0.0093)	0.1113 ^aD^ (0.0041)	0.1909 ^bB^ (0.0964)	0.1394 ^aC^ (0.0542)
Inosine (0.1 mM) + L-glutamine (20 mM)	T_min_	N.M.	0.0281 ^cA^ (0.0041)	0.0463 ^bB^ (0.0051)	0.0591 ^aB^* (0.0053)	N.M.	0.0117 ^aA^ (0.0021)
	T_opt_	N.M.	0.0460 ^bB^ (0.0017)	0.0524 ^bB^ (0.0032)	0.0623 ^aB^* (0.0056)	N.M.	0.0269 ^bA^ (0.0053)
	T_max_	N.M.	0.0917 ^aD^ (0.0066)	0.0861 ^aB^ (0.0062)	0.0692 ^aB^* (0.0043)	0.0353 ^C^ (0.0098)	0.0516 ^cA^* (0.0044)
Inosine (0.1 mM) + L-glycine (20 mM)	T_min_	N.M.	0.0329 ^cA^ (0.0042)	N.M.	0.0551 ^aB^ (0.0041)	0.0117 ^bC^ (0.0073)	0.0117 ^bA^ (0.0032)
	T_opt_	N.M.	0.0789 ^bD^ (0.0109)	N.M.	0.0613 ^aB^ (0.0063)	0.0382 ^aC^ (0.0072)	0.0532 ^aC^* (0.0063)
	T_max_	0.0488 ^C^ (0.0082)	0.1391 ^aE^* (0.0132)	N.M.	0.0544 ^aE^ (0.0046)	0.0491 ^aC^ (0.0081)	0.0733 ^aC^* (0.0184)

N.M.: not modeled. OD_600_ values did not decrease more than 15%.

**Table 4 foods-13-03434-t004:** Germination rate of non-activated and heat-treated spores produced at different temperatures in an ESL model at 25 °C (Appendix A), together with the percentage of germination at the end of the 4 h assay. Values in brackets correspond to standard deviations of the means calculated from three biological replicates. Different lowercase letters indicate statistically significant differences (*p* ≤ 0.05) among spore populations produced at different sporulation temperatures within each strain and heat treatment. Different capital letters indicate statistically significant differences (*p* ≤ 0.05) among heat treatments within each strain sporulated at a specific temperature.

Treatment	Strain	Sporulation Temperature	*k* (min^−1^)	R^2^	RMSE	Germination Efficiency (%) ^1^
Untreated	WSBC 10202	T_min_	N.M.	N.M.	N.M.	≤6.0 ^A^
		T_opt_	N.M.	N.M.	N.M.	≤6.0 ^A^
		T_max_	N.M.	N.M.	N.M.	≤6.0 ^A^
	WSBC 10204	T_min_	0.0210 ^aA^ (0.0071)	0.987	1.786	24.8 ^aA^ (0.2)
		T_opt_	N.M.	N.M.	N.M.	≤6.0 ^A^
		T_max_	0.0247 ^aA^ (0.0009)	0.923	1.390	25.1 ^aA^ (5.0)
	SC	T_min_	0.0240 ^bA^ (0.0110)	0.977	1.901	72.3 ^bA^ (1.3)
		T_opt_	0.0607 ^aA^ (0.0032)	0.944	2.642	92.5 ^aA^ (0.2)
		T_max_	0.0479 ^abA^ (0.0153)	0.937	1.804	69.9 ^bA^ (0.1)
80 °C, 10 min	WSBC 10202	T_min_	N.M.	N.M.	N.M.	≤6.0 ^A^
		T_opt_	N.M.	N.M.	N.M.	≤6.0 ^A^
		T_max_	N.M.	N.M.	N.M.	≤6.0 ^A^
	WSBC 10204	T_min_	0.0360 ^aB^ (0.0014)	0.9410	3.094	90.4 ^aB^ (4.0)
		T_opt_	0.0429 ^abA^ (0.0058)	0.9294	3.415	84.7 ^aB^ (5.3)
		T_max_	0.0502 ^bB^ (0.0054)	0.9413	2.719	82.6 ^aB^ (11.1)
	SC	T_min_	0.0043 ^bB^ (0.0029)	0.8949	5.849	30.6 ^cB^ (7.2)
		T_opt_	0.0621 ^aA^ (0.0086)	0.8087	4.809	82.3 ^aA^ (9.6)
		T_max_	0.0388 ^aA^ (0.0191)	0.8190	3.171	66.3 ^bA^ (3.5)
90 °C, 10 min	WSBC 10202	T_min_	N.D.	N.D.	N.D.	N.D.
		T_opt_	N.M.	N.M.	N.M.	21.8 ^B^ (11.7)
		T_max_	N.D.	N.D.	N.D.	N.D.
	WSBC 10204	T_min_	0.0303 ^aA^ (0.0013)	0.946	3.164	82.3 ^aB^ (5.9)
		T_opt_	0.0301 ^aB^ (0.0034)	0.925	3.605	92.6 ^aB^ (2.2)
		T_max_	0.0137 ^bC^ (0.0001)	0.921	4.238	91.1 ^aB^ (2.2)
	SC	T_min_	N.D.	N.D.	N.D.	N.D.
		T_opt_	N.D.	N.D.	N.D.	N.D.
		T_max_	N.D.	N.D.	N.D.	N.D.

N.M.: not modeled. OD_600_ values did not decrease more than 15%. N.D.: not determined. Germination was not tested because heat treatment caused >1.5 log reductions. ^1^ The lower limit of quantification was ≤6.0.

## Data Availability

The original contributions presented in the study are included in the article/Appendix A, further inquiries can be directed to the corresponding author.

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
