# Peer review of "Influence of Sporulation Temperature on Germination and Growth of B. weihenstephanensis Strains in Specific Nutrients and in an Extended Shelf-Life Refrigerated Matrix Under Commercial Pasteurization and Storage Conditions"

_foods, 2024, doi:10.3390/foods13213434_

Round 1

Reviewer 1 Report

Comments and Suggestions for Authors

First of all, contamination by heat-resistant bacteria in food is a current problem, and also how temperature affects the germination and growth of three strains of Bacillus, post pasteurization process. The details of my comments are as follows: 

1. There are different formats to express the international unit of measurement °C, they were highlighted in the PDF document in attachment.

2. In the introduction they talk about sporulation and spores in a very technical way, but if the authors want to reach a non-specialist public in microbiology it is necessary to clarify what a bacterial spore is, why it is produced and if they managed to count them they could have an image of these. 

3. Additional line 34 Clostridium botulinum

4. Line 122 This sporulation assay, at the location of Table 1, I find it difficult to understand why these results are in materials and methods.

5. Line 236. Idem 122, the title of Table 2 is too long, it is mixed up with methods. 

6. In Figure 1, the high standard deviations of the experiments are striking. 

7. The conclusion should be improved, usually there are no bibliographic references, besides it seems to be more a complement of the results than a conclusion which should be focused on the sporulation temperature, Effect of thermal activation and ESL Matrix.

Author Response

Please, see the attachment,
Thank you

Reviewer 2 Report

Comments and Suggestions for Authors

1. The authors' refrigerated temperature choices of 4°C and 8°C, 4 degrees is the temperature of a normal refrigerator freezer, what is the reason for the 8 degree choice?

2. Generally speaking, Bacillus cereus is not a cold-tolerant strain, but cold-tolerant Bacillus cereus still has the ability to grow below 10°C and can successfully colonize. It is recommended that the authors add an assessment of the production of metabolites (toxins) by B. cereus at low temperatures and the possible hazards.

3. The conclusions throughout the text are lengthy and it is recommended that the authors streamline them and some of them could be placed in the results section.

Comments on the Quality of English Language

The quality of English expression throughout the manuscript is good, with some phrases needing to be streamlined. 

Author Response

Please see the attachment,
Thank you
